# Dietary and Sexual Correlates of Gut Microbiota in the Japanese Gecko, *Gekko japonicus* (Schlegel, 1836)

**DOI:** 10.3390/ani13081365

**Published:** 2023-04-16

**Authors:** Xin-Ru Jiang, Ying-Yu Dai, Yu-Rong Wang, Kun Guo, Yu Du, Jian-Fang Gao, Long-Hui Lin, Peng Li, Hong Li, Xiang Ji, Yan-Fu Qu

**Affiliations:** 1College of Life Sciences, Nanjing Normal University, Nanjing 210023, China; 2Zhejiang Provincial Key Laboratory for Water Environment and Marine Biological Resources Protection, College of Life and Environmental Sciences, Wenzhou University, Wenzhou 325035, China; 3Hainan Key Laboratory of Herpetological Research, College of Fisheries and Life Science, Hainan Tropical Ocean University, Sanya 572022, China; 4Herpetological Research Center, College of Life and Environmental Sciences, Hangzhou Normal University, Hangzhou 311121, China

**Keywords:** ASVs, diet habit, *Gekko japonicus*, gut microbiota, sex

## Abstract

**Simple Summary:**

We used wild-caught Japanese geckos (*Gekko japonicus*) and captive conspecifics fed with mealworms and fruit flies to study their differences in gut microbial structure and composition and sexual correlates of gut microbiota. Gut microbial community richness and diversity were higher in mealworm-fed geckos than in wild geckos. The beta rather than alpha diversity of gut microbiota was sex dependent. From this study, we know the following. First, dietary and sexual correlates of gut microbiota are evident in *G*. *japonicus*. Second, with respect to the composition of gut microbiota, *G*. *japonicus* is more similar to the common leopard gecko than other reptilian taxa. Third, the diversity of gut microbiota is higher in geckos ingesting food with a higher chitin content.

**Abstract:**

Numerous studies have demonstrated that multiple intrinsic and extrinsic factors shape the structure and composition of gut microbiota in a host. The disorder of the gut microbiota may trigger various host diseases. Here, we collected fecal samples from wild-caught Japanese geckos (*Gekko japonicus*) and captive conspecifics fed with mealworms (mealworm-fed geckos) and fruit flies (fly-fed geckos), aiming to examine the dietary and sexual correlates of the gut microbiota. We used 16S rRNA gene sequencing technology to determine the composition of the gut microbiota. The dominant phyla with a mean relative abundance higher than 10% were Verrucomicrobiota, Bacteroidota, and Firmicutes. Gut microbial community richness and diversity were higher in mealworm-fed geckos than in wild geckos. Neither community evenness nor beta diversity of gut microbiota differed among wild, mealworm-fed, and fly-fed geckos. The beta rather than alpha diversity of gut microbiota was sex dependent. Based on the relative abundance of gut bacteria and their gene functions, we concluded that gut microbiota contributed more significantly to the host’s metabolic and immune functions. A higher diversity of gut microbiota in mealworm-fed geckos could result from higher chitin content in insects of the order Coleoptera. This study not only provides basic information about the gut microbiota of *G. japonicus* but also shows that gut microbiota correlates with dietary habits and sex in the species.

## 1. Introduction

The gut microbiota is known as the second genome of the host [1], encoding 10–100 times the number of genes in the host genome [2]. The gut microbiota plays a key role in host survival and adaptation, with its functions mainly manifested in a host’s life history [3], physiology [4], immune [5], growth [6], development [4], and behavior [7]. The gut microbiota can change rapidly in response to changes in the host’s environmental conditions and dietary habits [8], induce a host’s metabolic flexibility and phenotypic plasticity, and therefore enhance its ability to adapt to the environment [3]. Taxonomical shifts in gut bacterial communities in juvenile ostriches (*Struthio camelus*) coincide with the cessation of yolk absorption and co-occur with their dietary switch, and these shifts may help them adapt to dietary changes [6]. Short-chain fatty acids produced by the gut microbiota can maintain gut homeostasis [9]. Gut microbial dysbiosis can induce various host diseases and even threaten host survival [10].

The structure and diversity of gut microbiota are susceptible to numerous external and internal factors, including the host’s taxonomic category [11,12], sex [13], healthy status [14], age [6], dietary habit [15], and living environment [16]. These factors can substantially influence the composition, abundance, and diversity of gut bacterial communities. For example, captivity changes the gut microbial composition in a diverse array of terrestrial vertebrates, such as the crocodile lizard *Shinisaurus crocodilurus* [17], the northern grass lizard *Takydromus septentrionalis* [18], and the spectacled bear *Tremarctos ornatus* [19]. Qinghai toad-headed lizards (*Phrynocephalus vlangalii*) from the highest altitude population have the lowest gut microbial diversity [16]. In contrast, Glires mammals from high-latitude regions have a higher gut microbial diversity than their low-latitude conspecifics because the increased energy demands in cold and hypoxic environments cannot be met without increasing gut microbial diversity [20].

The gut microbial composition is largely host specific [12]. In invertebrates, the dominant gut microbial phyla are Tenericutes, Firmicutes, and Proteobacteria in snails [21,22] and Proteobacteria and Firmicute in insects [23]. In vertebrates, the dominant gut microbial phyla are Firmicutes and Bacteroidetes in amphibians [24], reptiles [25], and mammals [12] and Proteobacteria and Firmicutes in birds [26]. It is of great significance to explore the factors affecting gut microbiota and host–microbe symbiotic relationships. One widely accepted idea is that diet and host genetic status have a key role in shaping gut microbial structures [12,23].

Each microbial taxon has a functional role in the host gut. Bacteria of the phylum Bacteroidetes are the homeostasis cornerstone in a healthy gut and are involved in various functions, including gut–brain axis interactions, the immune system, and metabolic homeostasis [27]. A large number of genes of the phylum Firmicutes are clustered to encode the ABC-type sugar transport systems, and bacteria of this phylum are usually active in carbohydrate metabolism [28]. The ratio of Bacteroidetes to Firmicutes in relative abundance is correlated with diet, with higher ratios hinting at more efficient digestion [29]. The phylum Proteobacteria is regarded as a potential diagnostic signature of dysbiosis and risk of disease in human [28], but the ratio of Proteobacteria to Firmicutes and Bacteroidetes in relative abundance is correlated with bacterial stress tolerance under cold environments [30]. It is the role of gut microbiota that assists the host in adapting to a wide variety of diets and environments.

Reptiles are the first group of vertebrates that can truly live out of water on land, and their gut bacterial variation has therefore attracted much attention. As in other animal taxa, gut microbiota is affected by many factors in reptiles, including host genetic status [25], captivity [18], environment [16], and diet [17]. Previous studies on reptiles have shown that, for a given factor, it may affect the gut microbiota in some species but not in others. For instance, diet shapes the gut microbiota in *S. crocodilurus* [17] but not in the common water monitor *Varanus salvator* [31]. Compared with other reptile taxa, studies on gut microbiota in geckos have been limited, focusing only on the effects of fasting on gut microbiota in the common leopard gecko *Eublepharis macularius* [32,33] and the structure of gut microbiota in the common house gecko *Hemidactylus frenatus* [34]. Here, we used high-throughput sequencing to study the dietary and sexual correlates of the gut microbiota in the Japanese gecko *Gekko japonicus*. This gecko is a small-sized, oviparous species of the family Gekkonidae, occurring in the central and southeastern parts of China, Japan, and Korea. The gecko is a comparatively well-known lizard species in China, with data collected over the past few years covering a wide range of topics, such as genomics [35], temperature-dependent sex determination [36,37], molecular basis of character development [38], and microhabitat use [39].

## 2. Materials and Methods

### 2.1. Sample Collection

We used 49 adult geckos without any signs of disease (including ectoparasites) to conduct this study. All these geckos were collected on the Xianlin Campus of Nanjing Normal University (NNU), 25 (14♀♀ and 11♂♂) in June 2020 and 24 (11♀♀ and 13♂♂) in September 2020. Geckos collected in June were individually housed in 175 × 175 × 152 mm (length × width × height) plastic cages placed in a room where temperatures varied naturally. Of the 25 geckos, 13 (7♀♀ and 6♂♂; hereafter mealworm-fed geckos) were fed with mealworms (larvae of *Tenebrio molitor*) and 12 (7♀♀ and 5♂♂; hereafter fly-fed geckos) with fruit flies (*Drosophila melanogaster*), both for three months, during which period distilled water was available ad libitum. All facilities were disinfected by wiping 97% alcohol every other day. Mealworm- and fly-fed geckos always had free access to food sterilized with UV light for 1 h in advance to minimize the potential effects of bacteria in food on gut microbes. Geckos collected in mid-September (hereafter wild geckos) were individually housed in sterile 175 × 175 × 152 mm cages overnight, and then, fecal samples were collected. In September, we used light traps to collect insects at the sites where we collected geckos, thereby assessing prey items potentially available to geckos in the wild. Insects of the orders Lepidoptera and Diptera were the most abundant prey items potentially available to Japanese geckos on the Xianlin Campus of NNU (Table 1).

We put fecal samples collected from mealworm-fed, fly-fed, and wild geckos into sterile tubes, labeled these tubes, and then stored them at −20 °C for late DNA extraction. We released all geckos at their point of capture soon after the collection of fecal samples in mid-September. Geckos of different groups did not differ from each other in mean values for body mass (*H*_2,49_ = 1.95, *p* = 0.38) and snout-vent length (*H*_2,49_ = 2.62, *p* = 0.27). Our experimental procedures complied with laws on animal welfare and research in China and were approved by the Animal Research Ethics Committee of Nanjing Normal University (Permit No. IACUC 20200511).

### 2.2. DNA Extraction, PCR Amplification and Sequencing

We used the Mag-Bind Soil DNA Kit (Omega, Shanghai, China) to extract the microbial DNA from the fecal samples according to the manufacturer’s protocols. We used 2.0% agarose gel electrophoresis and a Qubit 3.0 DNA detection kit (Thermo Fisher Scientific, Waltham, MA, USA) to purify and quantify the DNA products, respectively. The bacterial V3–V4 region of the 16S rRNA gene was amplified using PCR with a 30 μL reaction system including 15 μL of 2× Hieff^®^ Robust PCR Master Mix (2×), 1 μL of each primer (10 μM), 20 ng of genomic DNA, and ddH_2_O. The universal primers 341F (5′-CCTACGGGNGGCWGCAG-3′) and 805R (5′-GACTACHVGGGTATCTAATCC-3′) were selected to perform the PCR reaction. The first round of PCR thermal cycling conditions was performed as follows: initial denaturation at 94 °C for 3 min, followed by 5 cycles of denaturation at 94 °C for 30 s, annealing at 45 °C for 20 s, and extension at 65 °C for 30 s. The other 20 cycles consisted of 94 °C for 20 s, 55 °C for 20 s, and 72 °C for 30 s, with a final extension at 72 °C for 5 min. In the second round, PCR products of the first round were used for amplification, and Illumina bridge PCR-compatible primers were introduced. The PCR reaction system was the same as in the first round. The thermal cycling conditions were as follows: denaturation at 95 °C for 3 min, followed by 5 cycles of denaturation a 94 °C for 20 s, at 55 °C for 20 s and 72 °C for 30 s, and a final extension at 72 °C for 5 min. Sequencing of the PCR-amplified products was conducted on Illumina MiSeq (San Diego, CA, USA).

### 2.3. Quality Control and Data Standardization

We imported the raw paired-end sequence into Quantitative Insights into Microbial Ecology 2 (QIIME2) using the manifest file and trimmed the primers [40]. We used DADA2 to filter and truncate low-quality reads and produce paired-end reads [41]. After quality control, these reads generated the raw amplicon sequence variants (ASV) with a minimum overlap of 12 bp. ASVs with a number of ASVs greater than 10 in at least two samples were retained for further analysis to avoid the effect of low read numbers on the results using QIIME2. The raw paired-end sequences were submitted to the National Genomics Data Center (NGDC) GSA database (accession number CRA007161).

We used QIIME2 to classify ASVs into organisms based on pre-formatted SILVA 138 SUU NR99 ASVs full-length reference sequences following the q2-fragment-classifier method in QIIME2. The sequencing depth for each sample was calculated using QIIME2 and visualized using R 4.0 [42]. We removed ASVs with a number less than 10 in only one sample for further analysis to avoid large partial sample deviations. The abundance information was standardized based on the sample with the least ASVs.

### 2.4. Estimation of Alpha and Beta Diversity

We used QIIME2 to calculate alpha diversity indexes, including community richness (observed species), community diversity (Shannon’s entropy index), and community evenness (Pielou’s evenness index). We used the Kruskal–Wallis *H* and Mann–Whitney *U* tests to examine whether alpha diversity indexes differed between (mealworm-fed, fly-fed, and wild) gecko groups and between sexes, respectively. Pairwise comparisons using the Wilcoxon rank sum test with continuity correction were performed when necessary. For beta diversity, we used principal coordinate analysis (PCoA) and permutational multivariate analysis of variance (Adonis) to show differences in microbial community structure among the gecko groups. Adonis was performed based on the Bray–Curtis distance with 999 permutations. Linear discriminant analysis of effect sizes (LEfSe) and linear discriminant analysis (LDA) were conducted to compare the microbial abundances from the phylum to genus levels based on a relative abundance higher than 1% [43]. The unique bacterial taxa were determined based on a log LDA score > 2 and *p* < 0.05. The Kruskal–Wallis *H* test was used to verify whether the bacteria detected by LDA had a higher relative abundance among the different diet × sex combinations.

### 2.5. Gene Function Predication

PICRUST2 (Phylogenetic Investigation of Communities by Reconstruction of Unobserved States, Nova Scotia, Canada) was used to explore the gene functions of all ASVs in gut microbiota based on the Kyoto Encyclopedia of Genes and Genomes (KEGG) database [44]. We allocated these gene functions to the corresponding KEGG pathways and obtained KEGG Orthology (KO) information for each gene function for the three KEGG pathways [45]. The relative abundance of these gene functions for each sample was calculated to assess the functional differences in gut microbiota among the different gecko groups. The LEfSe and LDA were performed to compare the relative abundance of KEGG gene functions from level 1 to level 3 based on a relative abundance higher than 1%. Only the gene functional category with a log LDA score > 2 and *p* < 0.05 was used in this analysis. The Kruskal–Wallis *H* test was used to verify whether the gene function detected by LDA had a higher relative abundance among the different diet × sex combinations. The unique and shared gene functions were visualized using a Venn diagram. All values were presented as mean ± standard error (SE), and the significance level was set at α < 0.05.

## 3. Results

We obtained 4,299,671 raw reads and 2,473,062 high-quality reads from the 49 fecal samples (Appendix A). The number of observed bacterial ASVs first increased with the increase in the number of sequences and then leveled out in each sample (Appendix A). We identified 976 bacterial ASVs, with 114–214 ASVs per sample (Appendix A). These ASVs could be allocated to 12 phyla, 19 classes, 49 orders, 83 families, and 168 genera.

The top four dominant bacterial phyla were Verrucomicrobiota (36.6 ± 3.5%), Bacteroidota (29.4 ± 2.2%), Firmicutes (18.9 ± 2.2%), and Proteobacteria (9.6 ± 2.3%) (Figure 1A). The dominant bacterial families with a relative abundance > 3% were Akkermansiaceae (35.3 ± 3.5%), Bacteroidaceae (18.0 ± 1.6%), Tannerellaceae (8.1 ± 1.0%), Enterobacteriaceae (6.2 ± 1.6%), Lachnospiraceae (4.4 ± 0.8%), and Clostridiaceae (4.3 ± 1.0%) (Figure 1B). The dominant genera with a relative abundance > 3% were *Akkermansia* (35.3 ± 3.5%), *Bacteroides* (18.0 ± 1.6%), *Parabacteroides* (5.7 ± 0.7%), and *Clostridium_sensu_stricto_1* (4.3 ± 1.0%) (Figure 1C).

### 3.1. Dietary and Sexual Correlates of Gut Microbiota

The Mann–Whitney test showed that none of the three alpha-diversity indexes differed between the sexes (all *p* > 0.05). Pooling data for both sexes, we found that community diversity (*H* = 7.80, *df* = 2, *p* = 0.02) and richness (*H* = 7.53, *df* = 2, *p* = 0.02) rather than community evenness (*H* = 5.93, *df* = 2, *p* = 0.05) differed among mealworm-fed, fly-fed, and wild geckos. Specifically, gut microbial community richness and gut microbial community diversity were significantly higher in mealworm-fed geckos than in wild geckos (Figure 2).

The PCoA based on the Bray–Curtis distance showed a significant separation of gut microbiota among six diet × sex combinations (Adonis: *r*^2^ = 0.15, *F*_5,43_ = 1.46, *p* = 0.006), with the first and second axes explaining 16.9% and 11.2% of the total variance, respectively (Figure 3A). We then divided the data into two groups for PCoA to analyze the similarity of gut microbes between the sexes and among the diet groups. The significant separation of gut microbiota was found only between the sexes (Adonis: *r*^2^ = 0.06, *F*_1,47_ = 2.89, *p* = 0.001; Figure 3A), rather than in different diet groups (Adonis: *r*^2^ = 0.05, *F*_2,46_ = 1.21, *p* = 0.174; Figure 3B). In addition, neither in males (Adonis: *r*^2^ = 0.09, *F*_2,21_ = 1.10, *p* = 0.32; Figure 3C) nor in females (Adonis: *r*^2^ = 0.09, *F*_2,24_ = 1.11, *p* = 0.30; Figure 3D) did gut microbiota differ among mealworm-fed, fly-fed, and wild geckos. LEfSe analysis showed significant differences in the unique gut microbiota among fly- and mealworm-fed females, mealworm-fed males, and wild males (Figure 4). Specifically, the unique bacteria families Desulfovibrionaceae and Marinifilaceae were found in mealworm-fed males, the families Eggerthellaceae and Caulobacteraceae were unique in wild males, the unique bacteria genera *Eggerthella*, *Bacteroides,* and *Odoribacter* were found in fly-fed females, and the families Erysipelatoclostridiaceae and Tannerellaceae and genera *Desulfovibrio* and *Clostridium_sensu_stricto_1* were unique in mealworm-fed females (Figure 4). The Kruskal–Wallis *H* test showed that the relative abundance of the above bacterial taxa, except for the family Tannerellaceae, differed significantly among different groups (Appendix A).

### 3.2. The Predicted Metagenomes

The predicted functions in gut microbiota were mainly involved in metabolism (80.8 ± 0.2%), genetic information processing (12.8 ± 0.2%), cellular processes (3.2 ± 0.1%), environmental information processing (2.4 ± 0.1%), organismal systems (0.4 ± 0.01%), and human diseases (0.32 ± 0.02%) at the first level (Figure 5A). The second KEGG category level was composed of 31 functions, among which the most abundant categories with a relative abundance > 5% in gut microbiota had functions associated with carbohydrate metabolism (15.0 ± 0.1%), metabolism of cofactors and vitamins (13.5 ± 0.2%), amino acid metabolism (12.2 ± 0.1%), metabolism of terpenoids and polyketides (8.9 ± 0.1%), glycan biosynthesis and metabolism (6.9 ± 0.2%), metabolism of other amino acids (6.8 ± 0.1%), lipid metabolism (6.1 ± 0.1%), replication and repair (5.9 ± 0.1%), and energy metabolism (5.3 ± 0.05%) (Figure 5B). Among 157 KEGG functions at the third level, those with a relative abundance > 2% were biosynthesis of ansamycins (3.7 ± 0.1%), other glycan degradation (2.7 ± 0.1%), biosynthesis of vancomycin group antibiotics (2.6 ± 0.1%), and valine, leucine, and isoleucine biosynthesis (2.1 ± 0.02%) (Figure 5C).

A total of 157 known KO functional genes were identified. Geckos in six diet × sex combinations shared 135 genes (Figure 5D). LEfSe analysis based on KOs revealed a unique gene function related to energy metabolism in fly-fed females (Figure 5E). In wild females, gene functions related to carbohydrate metabolism (Ko00010 and Ko00051) and environmental information processing and membrane transport were unique (Figure 5E). Gut microbial functions in fly-fed males had three unique functions related to metabolism (Ko00473, Ko01055, and biosynthesis of other secondary metabolites; Figure 5E). Gut microbial gene functions in mealworm-fed males were mainly associated with metabolism (Ko00340, Ko00720, Ko00790, and metabolism of cofactors and vitamins; Figure 5E). The Kruskal–Wallis H test showed that the relative abundance of the above unique gene functions had significant differences among different groups (Appendix A).

## 4. Discussion

At the phylum level, the dominant gut microbes in *G. japonicus* were Verrucomicrobiota, Bacteroidota, Firmicutes, and Proteobacteria (Figure 1A). This is consistent with what has been observed in the leopard gecko (*Eublepharis macularius*) [32] but differs from the results reported for other reptilian taxa, such as the Tokay gecko (*Gekko gecko*) [46]. For example, the dominant gut microbial phyla are Proteobacteria, Bacteroidetes, and Firmicutes in lizards [16,31,47] and snakes [48,49], Bacteroidetes and Firmicutes in turtles [25,50], and Fusobacteria, Proteobacteria, Firmicutes, and Bacteroidetes in crocodiles [51,52]. This indicates that the dominant gut microbial phyla differ among animal taxa. In fact, even among animals of the same evolutionary clade, their gut microbiota may differ significantly. For example, the dominant gut microbial phyla differ significantly between two species of turtles [25] and among four species of snakes [48] reared under the same conditions. This inconsistency between species provides evidence of the genetic correlates of gut microbiota in reptiles.

Taxonomically, all gut dominant genera and families in *G. japonicus* belong to the four dominant phyla mentioned above. The members of the phylum Verrucomicrobiota are correlated with mucin-degrading, glucose homeostasis, and inducing regulatory immunity [53], as well as reducing obesity risk [54]. Bacteria of the phylum Bacteroidota have functional roles in degrading the high molecular weight organic matter, activating T-cell mediated responses, and producing butyrate to maintain gut homeostasis [55]. Many studies have shown that bacteria of the phylum Firmicutes contribute to degrading complex carbohydrates of both plants and hosts [56]. Members of the phylum Proteobacteria are related to degrading and fermenting the complex sugars and producing vitamins for their hosts [57].

There is evidence that gut microbial compositions are closely correlated with food ingested by hosts [31] and with their sex [22]. In this study, mealworm-fed geckos had higher gut microbial community diversity and richness, although diet diversity was higher in wild geckos (Figure 2). Food diversity is not associated with gut bacterial alpha diversity in *G. japonicus*, which is similar to the findings demonstrated in *V. salvator* [31], the greylag goose *Anser anser* [58], and the plateau pika *Ochotona curzoniae* [59]. However, there are some species, such as the three-spine stickleback *Gasterosteus aculeatus*, the sea perch *Perca fluviatilis* [60], and the rice frog *Fejervarya limnocharis* [61], in which gut microbial alpha diversity is negatively correlated with diet diversity. Gut microbial alpha diversity did not differ between the sexes in *G. japonicus*, similar to the results reported for a wide range of vertebrates, including fish [62], amphibians [63], birds [58], and mammals [64]. However, sexual differences in gut microbial diversity do exist in many animals, including fish [65], birds [66], and mammals [67]. Taken together, the available data show that dietary and/or sexual correlates of host gut microbial alpha diversity are species- or taxon-specific.

Sex and diet shaped the beta diversity of the gut microbiota in *G. japonicus* (Figure 3 and Figure 4). However, PCoA showed that gut microbial structure differed only between sexes, but not among mealworm-fed, fly-fed, and wild geckos (Figure 3). LEfSe showed that gut bacterial relative abundance differed not only between the sexes but also among the three groups of geckos ingesting different prey items (Figure 4). Bacteria of the families Eggerthellaceae and Caulobacteraceae were enriched in wild males. Eggerthellaceae bacteria play an important role in the transformation of bioactive secondary plant compounds in human feces [68], and Caulobacteraceae bacteria actively metabolize linear alkylbenzene sulfonates in soil [69]. The enrichment of bacteria of the genera *Bacteroides*, *Eggerthella,* and *Odoribacter* in fly-fed females was correlated with metabolism [70], polysaccharide degradation [71], and immune [72], respectively. A higher relative abundance of Erysipelatoclostridiaceae at the family level, and *Desulfovibrio* and *Clostridium_sensu_stricto_1* at the genus level in mealworm-fed females was also enriched. Bacteria of Bacteroidales [73], *Desulfovibrio* [56], and *Clostridium_sensu_stricto_1* [74] could contribute to metabolism, and members of Erysipelatoclostridiaceae play a role in immunity in the host gut [75]. Therefore, the differences in the relative abundance of gut microbiota may contribute more to metabolic and immune functions in the gecko.

The putative gut microbial functions in *G. japonicus* were mainly related to metabolism at the first function level with a relative abundance > 80%, the metabolism-related function, replication, and repair at the second level, antibiotic, and partial amino acid biosynthesis, and other glycan degradation at the third level with higher relative abundances (Figure 5). Putative gut microbial functions in most animals are closely related to metabolism, including fish [62], amphibians [76], reptiles [18], birds [77], and mammals [19]. Therefore, the gut microbiota plays an important role in host energy metabolism. This is also evidenced by the enrichment of putative gene functions with high relative abundance in different diet × sex combinations in *G. japonicus* (Figure 5). For example, a higher relative abundance of putative gene functions related to metabolism were enriched in all male geckos, fly-fed females, and wild female geckos.

Prey items potentially available for the Nanjing population of *G. japonicus* in September consisted of insects of the orders Lepidoptera and Diptera (Table 1). Insects mainly contain protein (30–70% of dry mass), fat (~35% of dry mass), minerals, and vitamins [78], and can modulate the gut microbiota and improve host health status [79]. The fruit flies and mealworms used in this study belong to the orders Diptera and Coleoptera, respectively. This might be the main reason why the gut microbiome of fly-fed geckos was closer to that of wild geckos. However, mealworm-fed geckos fed on mealworms containing more chitin. Chitin is one of the most abundant biopolymers in nature [79] and can restore the compositional balance of the bacterial community [78,80,81]. In this study, more diverse gut bacteria in mealworm-fed geckos might have resulted from abundant chitin in diets. Therefore, the gut bacterial alpha diversity in *G. japonicus* might be correlated with the type of insect diet.

## 5. Conclusions

Gut microbial community diversity and richness, rather than community evenness, differed between mealworm-fed geckos and wild conspecifics. More specifically, gut microbial community richness and diversity were significantly higher in mealworm-fed geckos than in wild geckos. None of the above three alpha-diversity indexes differed between the sexes. There was a significant separation of gut microbiota between the sexes. Such a separation did not exist among geckos of the same sex ingesting different prey items. The relative abundance of unique gut bacteria and gene functions differed among different diet × sex combinations. Our study is the first to demonstrate dietary and sexual correlates of gut microbiota in geckos.

## Figures and Tables

**Figure 1 animals-13-01365-f001:**
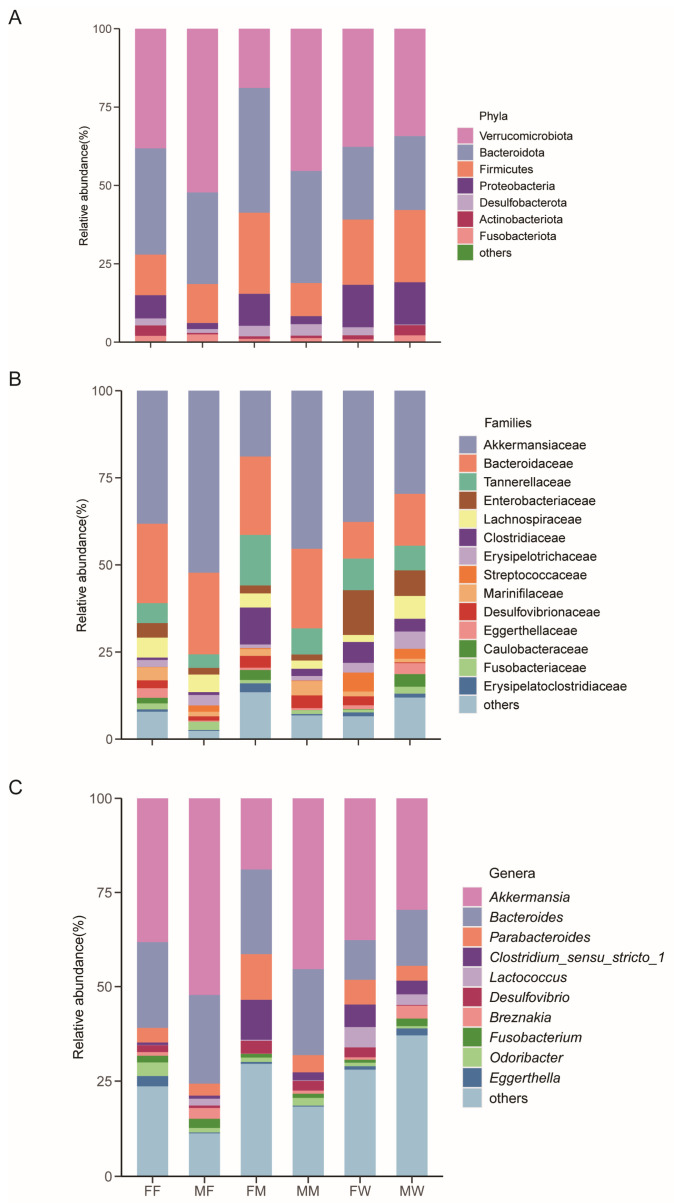
The relative abundance of the gut microbiota in each gecko group at the phylum (**A**), family (**B**), and genus (**C**) levels. Each color in a plot represents a taxonomic group, the name of which is shown on the right side of the plot. The color for ‘others’ indicates all other phyla (**A**), families (**B**), or genera (**C**) combined, of which the names are not listed in each plot. FF: fly-fed females; MF: fly-fed males; FM; mealworm-fed females; MM: mealworm-fed males; FW: wild females; MW: wild males.

**Figure 2 animals-13-01365-f002:**
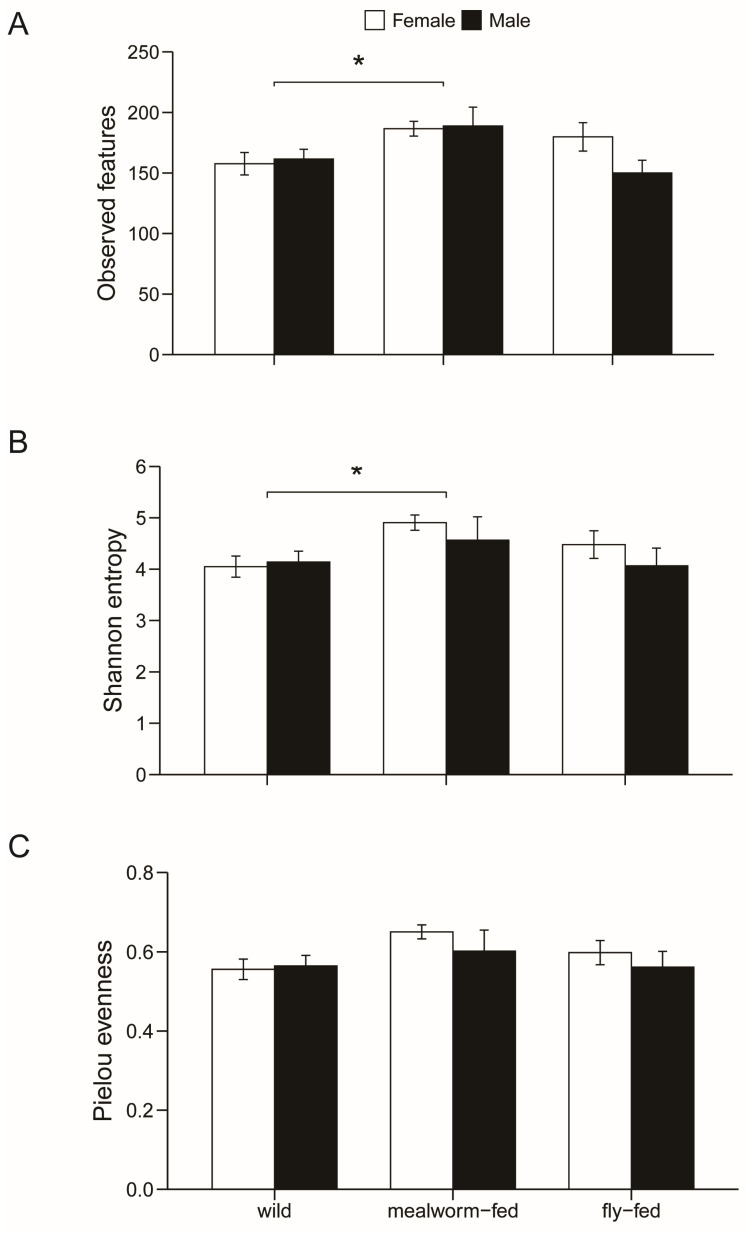
Mean values (±SE) for three alpha-diversity indexes of gut microbiota in six diet × sex combinations of fecal samples, including observed species (**A**), Shannon’s entropy index (**B**) and Pielou’s evenness index (**C**). Asterisks (*) indicates a significance level of *p* < 0.05.

**Figure 3 animals-13-01365-f003:**
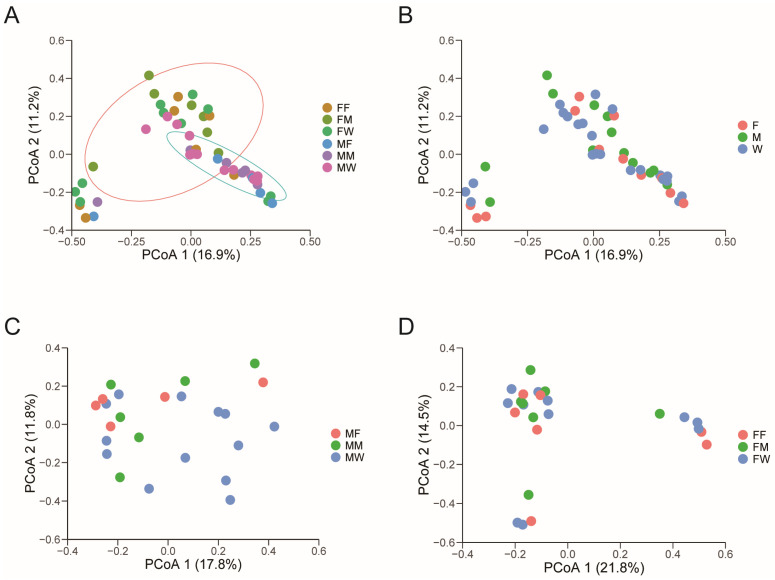
Gut microbial diversity in six diet × sex combinations of fecal samples (**A**), fecal samples from three different diet groups (**B**), male (**C**) and female (**D**) geckos ingesting different prey items. Principal coordinates analysis of the Bray–Curtis distance matrix for bacterial community diversity. See Figure 1 for the definition of each combination (group).

**Figure 4 animals-13-01365-f004:**
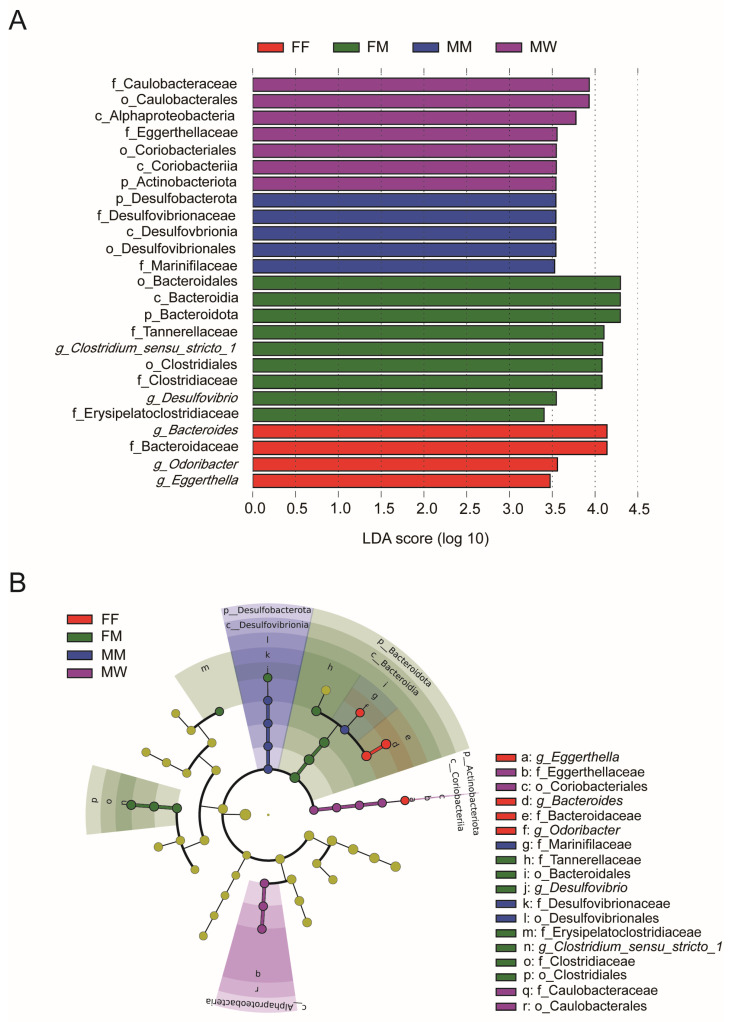
Differences in gut microbiota among the four groups were determined by LEfSe (**A**). LDA scores reflect the differences in relative abundance among four diet × sex combinations (groups) (**B**). See Figure 1 for the definition of each combination. The letters “o”, “f”, and “g” indicate order, family, and genus, respectively.

**Figure 5 animals-13-01365-f005:**
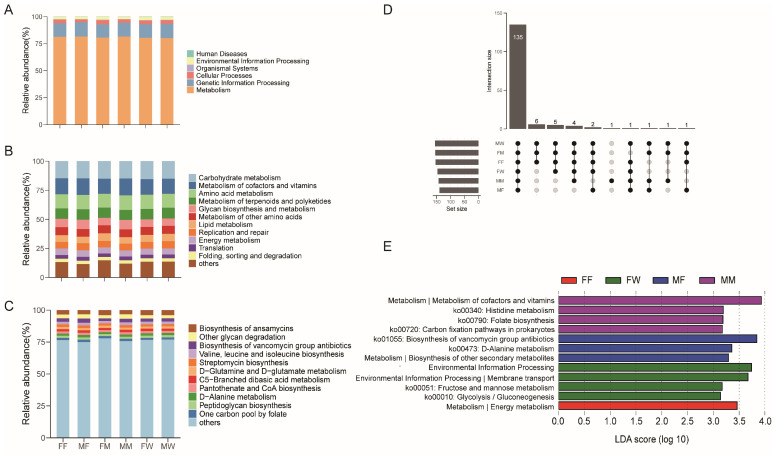
The relative abundance of gene functional categories based on 16S RNA in the gut microbiota at top (**A**), second (**B**), and third (**C**) levels, and the Venn diagram of functional gene among six diet × sex combinations (groups) (**D**). LDA scores reflect the differences in relative abundance among four diet × sex combinations (**E**). Each color in plots (**A**–**C**) indicates a gene function. Detailed descriptions are shown on the right side of each plot. The colors for the others in plots (**B**,**C**) indicate all other gene functions not listed in these two plots. See Figure 1 for the definition of each combination.

**Table 1 animals-13-01365-t001:** Prey items potentially available to Japanese geckos in the wild.

Abundance of Prey Items ^1^	Order
Numerous (>500)	Lepidoptera, Diptera
More (between 100 and 500)	Coleoptera, Hemiptera
Medium (between 50 and 100)	Hymenoptera, Ephemeroptera, Trichoptera
Fewer (between 10 and 50)	Orthoptera, Mantodea, Neuroptera, Megaloptera, Thysanoptera, Plecoptera, Blattodea
Least (<10)	Dermaptera, Odonata, Corrodentia, Rhaphidioptera

^1^ The abundance of prey items is sorted by the number of insects found in the light trap.

## Data Availability

All 16S rRNA gene sequences obtained in this study have been deposited in the National Genomics Data Center (NGDC) GSA database (accession number CRA007161).

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
