# Peer review of "Dietary and Sexual Correlates of Gut Microbiota in the Japanese Gecko, Gekko japonicus (Schlegel, 1836)"

_animals, 2023, doi:10.3390/ani13081365_

Round 1

Reviewer 1 Report

I read the paper and find it is very interesting paper about the structure and composition of gut microbiota analyzed by diets in Japanese geckos. The paper is well writing and good structure. I recommend the paper can be accepted after several minor revisions.

1. delete "for examples" in introduction.

2. On the contrary, Glires mammals from high-latitude regions have a higher gut microbial diversity than their low-latitude conspecifics, because the increased energy demands in cold and hypoxic environments cannot be met without increasing gut microbial diversity [20]. 

This example should be replaced by lizard example.

3. In the introduction, it is better to use past tense of English writing, especially in citing the references.

4. The species name should be italic "Tenebrio molitor and L Drosophila melanogaster"

5. described in PCR Primers and and llumina bridge PCR compatible primers. It  is better to add the reference of primers.

6. among mealworm-fed, fly-fed and wild geckos. It can be added one figure to compare those.

7. check the format of references. eg. Qin, J.-J.; Li, R.-Q.;

Reviewer 2 Report

Here the authors explore whether the structure and composition of gut microbiota are influenced by the host sex and diet. Gut microbiota is affected by many factors in reptiles, but compared with other reptile taxa, studies on gut microbiota in geckos have been limited. So this paper could provide new information for the study of gecko in reptiles. This paper is based on sufficient statistics and detailed charts. It mainly describes the bacterial abundance and gene function, alpha and beta diversity. And this paper can explain authors’ questions clearly and comprehensively.

However, in my opinion some details need to be clarified before publication. The detailed comments are as follows.

Abstract

lines 27-29: From title”3.2. Dietary and Sexual Correlates of Gut Microbiota”, gut microbial community richness and gut microbial community diversity were significantly higher in mealworm-fed geckos than in wild geckos, and were not significantly higher than in fly-fed geckos. So is it written wrong here?

lines 29-30: Why to use the neither... nor.... here? “Alpha diversity” differences were described above. Or maybe the “Alpha diversity” here says the “community evenness”?

Materials and Methods

lines 162-163: Are there any restrictions on how ASVs were distributed in the sample?

lines 224-228: The corresponding image in “different diet groups” is in Figure 3D; The corresponding image in “males and females ” are in Figure 3B and 3C.

lines 231: “Desulfovibrioria” should spell into “Desulfovibrionaceae”.

lines 283: From figue5D, Geckos in six diet ´ sex combinations shared 135 genes rather than 136.

Figure legends

Fig.1: Missing a group name abbreviation and note the use of semicolons and colons.The correct ones should be as follows. FF: fly-fed females; MF: fly-fed males; FM: mealworm-fed females; MM: mealworm-fed males; FW: wild females; MW: wild males.

Fig.3: The serial number error. The correct ones should be as follows. fecal samples from three different diet group (D), male (B) and female (C) geckos ingesting different prey items.

Reviewer 3 Report

The authors examined the fecal microbiomes of wild-collected Gekko japonicus sampled the day after capture or held for three months and fed either experimental mealworm or fly diets. They used 16s sequencing with QIIME workflows to analyze fecal microbial composition, finding more richness/diversity in mealworm-fed geckos compared to the natural diet group. I am not entirely clear on the beta-diversity results—I believe there were no differences in microbiome composition found between males and females of the same treatments, but that sex by treatment interactions showed distinct microbial communities between each of two pairs: females fed mealworms or flies, males fed mealworms or on a natural diet. The authors proceeded to use PICRUSt2 to predict functional capabilities of the gut microbes based upon taxonomic identification assessed from 16s sequences. They broadly categorize potential microbial functions and use these predictions to hypothesize differences in microbial contributions with respect to diet and sex. Their conclusions lean in support phylosymbiosis, specifically that G. japonicus have very similar fecal microbiomes to another gecko compared to lizards in general. They focus on the role of the high chitin content of mealworm diets in shaping the microbiome function and make mention of sex differences.

I think there are interesting and important aspects of this study, including closing some of the gap in the literature on gecko biology and conducting manipulative experiments with diet and microbiome. I appreciate the inclusion of sex as a fixed variable and the appropriate design of considering the interaction of diet and sex in their analyses. I am intrigued by the inclusion of functional considerations of the gut microbiome, but also have reservations about the conclusions that can be drawn from PICRUSt analyses in this context.

Major comments and questions:

1.     Did the authors sequence the microbiota of the food items? I see that the experimental diets were put under a UV light, but it’s not clear to me that the microbiota were part of the lizards’ consistent microbiome and not passing through as a component of the prey.

2.     Page 7. The results section for beta diversity was difficult to follow. Specifically which analyses belonged to which comparisons, and thus tracking differences was unclear (e.g. sex overall, sex by treatment, etc.). Additionally, the markers for figure 3 are not visually accessible. Consider going higher contrast and varying shapes (larger could also be good!).

3.     My understanding of PICRUSt analyses is that they can be used to form hypotheses based on ASV presences, but that due to the wide potential for diversity in the microbial genome outside of the 16s v3-4 region, matching these taxonomic categories back to functional ones is risky. I can appreciate that working at the phylum level and making predictions about metabolic vs. immune potential functions is much more appropriate than pinpointing specific functions, but that there is still quite a bit of variability between these types of results and metagenome analyses. I would be more comfortable with the functional results being clearly framed as hypotheses to follow up on, whereas in the discussion they are presented as more conclusive.

Minor comments and questions:

1.     The results aren’t especially clear from the simple summary.

2.     Page 2, end of the first full paragraph: “on the contrary” conveys the incorrect meaning. Consider replacing with “conversely” and acknowledging the difference in physiological needs between ectotherms and endotherms (i.e. zone of metabolic regulation)

3.     Page 2, third full paragraph: terms like “slim figure”, “healthier”, and “obesity” are such value-laden terms that are nigh impossible to separate from human contexts. While I think it’s relevant to mention the bacteriodetes:firmicutes ratio in the context of the human work it’s been most studied in, consider using more neutral (and applicable) wording like mentioning energy acquisition and storage.

4.     The introduction set up diet and other functions well in terms of their interplay with microbiomes, but sex felt more tacked on. Integrate sex a bit more into the background information? I appreciate the careful use of sex not being conflated with gender (although, I think a figure legend did inappropriately use gender).

5.      Fig 1. Especially with the layout of the review version, there’s space to make the taxa names larger. They’re so small! Similarly with the other figures.

6.     Page 6. Are these results with sex pooled?

7.     Fig. 2. I believe the error bars are SE, correct? I’d appreciate having that mentioned in the figure legend.

8.     Fig. 2. Can “wild”, “mealworm-fed”, and “fly-fed” be written out on the axis/axes?

9.     Page 9, discussion. What other geckos in addition to Leopard Geckos are there microbiome data for? How do those fit in with reptiles?

10.  Page 10, paragraph one. “Healthy gut” is really vague—consider replacing with upkeep of colonocytes, energy source for enterocytes, etc.

11.  Page 10, paragraph two. While I believe that the wild geckos had a more varied diet, it isn’t explicitly stated elsewhere. Did you flush any stomachs? Identify prey in feces? Figure 2 also doesn’t appear to be related to dietary diversity.

12.  Page 11, continuing paragraph. Wouldn’t an animal feeding in general contribute positively to its health status? I’m unclear what this means here.

13.  Page 11, conclusions. The sex aspect feels somewhat abrupt and doesn’t seem to tie back to the biology of the gecko. Why would or wouldn’t there be sex differences in the microbiome?

Best regards
